# Turning Research of Additive Laser Molten Stainless Steel 316L Obtained by 3D Printing

**DOI:** 10.3390/ma12010182

**Published:** 2019-01-07

**Authors:** Grzegorz Struzikiewicz, Wojciech Zębala, Andrzej Matras, Magdalena Machno, Łukasz Ślusarczyk, Sebastian Hichert, Fabian Laufer

**Affiliations:** 1Production Engineering Institute, Mechanical Faculty, Cracow University of Technology, 31-155 Kraków, Poland; zebala@mech.pk.edu.pl (W.Z.); amatras@mech.pk.edu.pl (A.M.); magdalena.machno@mech.pk.edu.pl (M.M.); slusarczyk@mech.pk.edu.pl (Ł.Ś.); 2Institute of Manufacturing Technology and Quality Management (IFQ), Faculty of Mechanical Engineering, Otto von Guericke University Magdeburg, 39106 Magdeburg, Germany; sebastian.hichert@ovgu.de (S.H.); fabian.laufer@ovgu.de (F.L.)

**Keywords:** 316L stainless steel, sintered materials, turning, SLM

## Abstract

This paper presents the characteristic of 316L steel turning obtained by 3D printing. The analysis of the influence of turning data on the components of the total cutting force, surface roughness and the maximum temperature values in the cutting zone are presented. The form of chips obtained in the machining process was also analyzed. Statistical analysis of the test results was developed using the Taguchi method.

## 1. Introduction

In recent years, an intensive Additive Manufacturing (AM) industry has developed, due to possibilities of manufacturing of very complex structures inside parts and work pieces. This innovative technology is often determined as “3D printing”. The technology, in contrast to the conventional (subtractive) method of “top-down” machining processes (SM) which removes material, creates parts via a “bottom-up” process. In AM, the elements are created layer-by-layer by using a computer-controlled laser beam source. This allows fabrication of complex shapes inside of the parts, which are difficult to obtain by using molding or conventional manufacturing techniques [1,2,3]. AM technologies are used in several applications such as turbine blade manufacturing in aerospace engineering, prosthesis and medical implants (in the medical industry), and die casting molds, valves, heat exchangers, manifolds and collectors. One of the main challenges of this technology is the characterization and prediction of manufactured structures and their connection with selected fabrication settings [4,5,6,7].

Both subtractive and additive manufacturing have several advantages and disadvantages. In practice, parts made in additive technology are a replacement for casting technology. The big advantages of the additive manufacturing rely basically on no restrictions to building complex shapes and ability to produce several units at the same time. On the other hand, the disadvantage is undoubtedly the lower surface quality and the dimensional–shape accuracy compared to parts made with the use of machining. The material obtained with 3D printing technology shows porosity and areas with different consistency of material. The characteristics of the parts obtained in the AM technology is the layered structure of the material and necessity to decrease the size of stresses within the material caused by a poor heat distribution [8,9].

Geometrically simple objects are easier to machine by SM. A more complex tool path is generated if the geometry becomes more complex and this can be difficult to produce with a milling machine even with five or more axes.

An interesting idea to improve the quality of the manufactured parts is the use of hybrid processing, previously analyzing whether it is better to use more thin layers during AM production or fewer thick layers (then worse quality of the item) but with the need of additional mechanical machining AM + SM [10].

Selective Laser Melting (SLM) is the most widespread method of metal additive manufacturing that allows laser fusion of metal powders (with low granulation 15–45 μm), such as AISI 316L and PH 17-4 stainless steels, Maraging steels, Inconel 625 and 718, Al-Si alloys, CoCr and Ti6Al4V [1,4,11]. SLM involves a powder bed fusion process where parts are produced by melting a thin coat of powder layer-by-layer by applying a rastered laser beam controlled directly from a computer aided design model (CAD). For the SLM-process, the build chamber is flooded with an inert gas, such as argon (Ar), or a process gas, such as nitrogen (N_2_), to prevent oxidation on the powder surface [1,6]. Additionally, the SLM process allows the fabrication of almost fully dense metal parts with mechanical properties comparable to components made by conventional routes. SLM enables a high relative density (even for arbitrary complex structures), low porosity of the manufactured elements and high material utilization rates. SLM enables the fabrication of parts without the need of tools and prototypes. In addition, for the production of small element sizes, there is no increase in costs (in contrary to the mold-based technologies). However, the quality of SLM produced parts strongly depends on the laser processing parameters, building chamber atmosphere, powder bed preheating, and especially on the powder feedstock characteristics [11,12,13,14,15,16,17]. Unfortunately, the parts fabricated by using the SLM process, characterize decreased dimensional accuracy and surface quality (with the dimensional tolerance of 40–80 μm). Therefore, post-processing by machining (finishing or grinding) is often necessary in order to achieve the suitable surface conditions for its applications [18,19].

Several alloys are currently used for SLM. However, the AISI 316L austenitic stainless steel is frequently employed due to its combination of good mechanical properties, excellent corrosion resistance and good SLM process compatibility [4,12]. Researchers have focused mainly on applying process parameters for SLM to improve the geometrical and microstructural properties of manufactured components (such as surface roughness and surface integrity, high density and residual stress of AISI 316L) [13,20,21]. The effect of LSM parameters such as laser power, scanning velocity and scanning direction are correlated with the mechanical properties of the fabricated parts [22,23,24]. For example, the lower laser power *P* = 100 W causes the fabricated sample to be characterized by more pores and lower hardness [23]. However, the laser power *P* = 200 W and the high scan speed *v_scan_* = 400 mm/s yield a low temperature during process SLM, which gives poor wettability and micropores appearing in the structure [24]. Additively, the laser power increase and or scanning velocity decrease contribute to the increase of energy density, which impact on: lower porosity, bigger melt pool, and stronger bonding between pools [22]. The quality of fabricated part depends on these properties, which can affect the post-process machining. 

Additively, the parameters of SLM such as energy input effects spatter behavior, which is generated during the process. The formed spatter influences the part structure, which can be significant during added post-process machining [25]. The distribution state and particle size of nanoscale TiC reinforcements in the structure of molten part affect the micro-hardness and wear rates [26]. The increase of the volume content of TiC causes an increase of hardness and decrease of coefficient of friction and wear rate [27].

Previous studies to a lesser extent have presented the machining of external surfaces of AISI 316L fabricated via SLM. The impact of machining data on the surface quality (surface roughness or surface integrity) plays an important role [28,29,30,31]. Therefore, there is a need to analyze the cutting process of metals obtained by additive technology, SLM in our case.

We determined the influence of parameters cutting data on the molten laser stainless steel 316L on the surface finish, cutting forces and temperatures values in the cutting zone. A machining analysis of a specially designed sintered turning ring was carried out.

## 2. Materials and Methods

### 2.1. Design of Experiment

The external surface of the test sample was subjected to the longitudinal turning. The 3D samples made of 316L stainless steel (with grain size in range 23–48 µm) were fabricated with SLM/LMF technique using a TRUMPF TruPrint 1000 3D Laser Metal Fusion machine (Ditzingen, Germany). Figure 1 presents a ring-shaped sample and its dimensions. The SLM process parameters are characterized in Table 1.

The prepared shape of the test stand enables the installation on a specially prepared attachment bolt. The test position and applied cutting tool, produced by ISCAR company (Tefen, Israel), the cutting insert, DCGT 11T302-AS IC20; and the tool holder, SDJCR2020K11) [32] are shown in Figure 2.

During the investigations, measurements of surface roughness, surface topography, cutting forces, micro-hardness and temperature in the cutting zone were performed. Furthermore, the created chips were classified (advantages, disadvantages, acceptable or unacceptable). The turning tests of the sample were carried out on a Masterturn 400 lathe, equipped with a special prepared measurement system that enables the measurement of cutting forces and temperature values in the machining area. The cutting forces were measured and recorded by applying the dynamometer 9257B and the amplifier 5070B produced by Kistler company (Winterthur, Switzerland). The cutting forces waveforms were converted by analog to digital converter and analyzed using DynoWare software (Version 2825A, Kistler Group, Winterthur, Switzerland). Sampling frequency was 1 kHz and measuring time was 10 s. The measurements of surface roughness and topography were performed using Talysurf Intra 50 profilometer produced by Taylor Hobson company (Leicester, UK). The microhardness was measured using Micro-Vickers HM-112 tester produced by Mitutoyo company (Kawasaki, Japan). After the investigations, the average hardness value of the external surface was determined. The measurements were performed along the radius of the workpiece over a distance of 0.5 mm. The analysis of the results did not show any changes in the hardness value, which was HRC_mean_ = 43 (HRC—Rockwell Hardness scale C).

### 2.2. Material

AISI 316L stainless steel is specified as X2CrNiMo17-12-2/1.4404 according to European standard and includes the austenitic structure of stainless steel. The chemical composition and main mechanical properties of this material are presented in Table 2 and Table 3, respectively. This stainless steel is used for fabricating parts working in salt water condition; for chemical, paper, and food industries among others; and for architectural elements. Additionally, due to anticorrosive properties, austenitic stainless steels are commonly used. The addition of Molybdenum (Mo) in chemical composition of the stainless steel contributes to increase acetic and sulfuric acid resistance.

### 2.3. Experimental Details

The experimental research was performed according to the Taguchi method [33]. The purpose of the tests was to examine the impact of some machining data, such as feed rate *f* and cutting speed *v_c_*, on:component values of the total cutting force *F*, main force *F_c_*, feed force *F_f_* and thrust force *F_p_*;surface roughness parameter values *Ra* and *Rz*; andmaximum temperature value *T_max_* in machining zone.

Cutting depth *a_p_* = 0.5 mm and cutting edge radius *r_ε_* = 0.2 mm were assumed. The cutting data values of the turning experiment are presented in Table 4. To statistically fit the experimental data, the polynomial was selected.

The strategy of factor analysis S/N (signal to noise) was determined as “smaller-is-better” according to following formula:(1)$$S/N=-10\cdot \log\left(\frac{1}{n}\sum_{i=1}^{n}y_{i}^{2}\right),$$
where: *y_i_* is the respective characteristic and *n* is the number of observations.

## 3. Results Analysis of Cutting Forces Measurements

In Table 5, the experimental results of the components of the total cutting force with standard deviations (*Std. Dev*.) are shown. The cutting forces, such as *F_c_*, *F_f_*, and *F_p_*, are presented as their average values. The impact of feed rate *f* on the cutting forces values and the total cutting force *F*, are presented in Figure 3.

The results show that the feed rate *f* significantly affects the total cutting force *F*. In the case of the longitudinal force *F_c_*, the increase of the feed rate values causes a stable increase of the force with constant increments. A threefold increase of feed rate, from *f* = 0.07 mm/rev to *f* = 0.211 mm/rev, contributes to the cutting force increase *F_c_* of about 120–150 N in relation to the cutting speed. It was observed that, when applying the cutting speed *v_c_* = 100 mm/min for feed rate *f* > 0.15 mm/rev, the increment of thrust force *F_p_* decreases and the increment of feed force *F_f_* increases. This result indicates changes in the direction of the forces (*F_f_* and *F_p_*) in the case of using higher cutting speed values. The analysis of the results shows a decrease of about 10% of the cutting force *F_c_* when using the higher cutting speed (100 m/min vs. 60 m/min). The obtained results of *S/N* parameters and its values for the cutting forces (*F_c_*, *F_f_*, and *F_p_*) are presented in Table 6. In Figure 4, the impact of cutting data on the cutting forces is shown.

The analysis of results confirms that the feed rate mainly affects the values of the cutting forces *F_c_*, *F_f_* and *F_p_*. The cutting speed increase causes a decrease of values for all components of the total cutting force. Table 7, Table 8 and Table 9 show the ANOVA regression analysis results of the components for the total cutting force (where: DF—degrees of freedom, Seq SS—sums of squares, Adj SS—adjusted sums of squares, Adj MS—adjusted means squares).

*F_c_*(*f*, *v_c_*), *F_f_*(*f*, *v_c_*) and *F_p_*(*f*, *v_c_*) are described by Equations (2)–(4):(2)$$F_{c}\left(f,v_{c}\right)=181.723+65.66f-8.75v_{c}+17.45f^{2}+1.71fv_{c},$$(3)$$F_{f}\left(f,v_{c}\right)=88.573+14.275f-0.295v_{c}+784.26f^{2}-0.68fv_{c},$$(4)$$F_{p}\left(f,v_{c}\right)=43.77+213.58f+0.098v_{c}+1051.21f^{2}-2.98fv_{c},$$

## 4. Results Analysis of Surface Roughness

Table 10 presents the results of measured surface roughness *Ra* and *Rz*. The examples of topographies and profiles of the parts surface are shown in Table 11 (Trial 1 for *f_min_* and Trial 7 for *f_max_*, *v_c_* = 60 m/min).

Figure 5 presents the impact of the feed rate on *Ra* and *Rz* roughness parameters, for values of the cutting speed *v_c_* = 60 m/min and *v_c_* = 100 m/min. Table 12 shows the *S/N* factor results and the average values of surface roughness parameters (*Ra* and *Rz*, respectively).

Presented analysis of the relations in Figure 5 shows that values of surface roughness parameters are proportional to feed rate values. In all cases, higher values of surface roughness *Ra* and *Rz* are obtained with higher values of cutting speed, during changes from *v_c_* = 60 m/min to *v_c_* = 100 m/min. Moreover, it was observed that a higher dispersion of the measured values for *v_c_* = 100 m/min is accrued.

The graphical representations of the impact of the surface roughness parameters *Ra* and *Rz* on the cutting data and *S/N* factor are presented in Figure 6.

The cutting feed increase causes the increase of surface roughness *Ra* and *Rz*. Results analysis presented in Figure 6 additively confirms the most significant impact of feed rate on the cutting forces *F_c_*, *F_f_* and *F_p_*. Table 13 and Table 14 present the ANOVA regression analysis results for each roughness parameter. *Ra*(*f*, *v_c_*) and *Rz*(*f*, *v_c_*) are described by Equations (5) and (6).

(5)$$Ra\left(f,v_{c}\right)=-0.142+10.05f+0.0035v_{c}-1.28f^{2}+0.1fv_{c},$$

(6)$$Rz\left(f,v_{c}\right)=-0.78+60.46f-0.062v_{c}-131.24f^{2}+0.335fv_{c},$$

Figure 7 shows photographs of the obtained chips for selected compositions of the experimental design (Trial 1 for *f_min_* and Trial 7 for *f_max_*). During the experimental research, the classification of created chips was performed. A three-step scale was adopted: “+”, advantageous chips; “−“, disadvantageous chips; and “0”, unacceptable chips. In all experimental tests, unacceptable chips were obtained (long, tangled, and spiral).

## 5. Results Analysis of Temperature in Cutting Zone

The temperature measurements were performed using a FLIR SC 620 thermal camera (FLIR Systems, Wilsonville, OR, USA)which was installed above the cutting zone and connected to a computer. ThermaCam Researcher Pro 2.9 (FLIR Systems, Wilsonville, OR, USA) was used for acquisition and analysis of the recorded thermograms. Two-second sequences of a stable phase of machining process (30 frames per second) were recorded and the maximum temperature *Tmax* that existed in the cutting area was obtained. The main errors during temperature measurements are the emissivity factor and reflections. In our case, the emissivity factor was 0.98. The configuration parameters of the thermal camera are presented in Table 15.

Figure 8 presents the thermal vision of the cutting zone and selected thermograms of tests for *v_c_* = 60 m/min *f_min_* = 0.07 mm/rev and *f_max_* = 0.211 mm/rev.

Based on the obtained experimental research using the Taguchi methods, the ANOVA regression analysis was performed. Table 16 and Table 17 present obtained results of statistical analysis and values of variance for average analysis.

The polynomial *T_max_* (*f, v_c_*) is described by Equation (7).
(7)$$T_{max}\left(f,v_{c}\right)=245-625f+0.13v_{c}+2245.59f^{2}-2.53fv_{c},$$

The impact of the cutting data on the values of the maximum temperature in the cutting zone is shown in Figure 9.

During the experimental research, the camera was installed perpendicular to the cutting zone and recorded the flown chip on the rake face of the cutting insert. It had the most impact on the recorded value of the temperature. The feed rate increase contributes to a decrease of the maximum temperature recorded by the thermal camera. The section of the cutting layer and chip thickness increase with a feed increase. Further, the part of generated heat flux on the junction chip and cutting edge spreads in more material volume. The cutting speed increase causes a decrease of the temperature value in the cutting zone. It can result from the shorter contact time between the chip and the cutting edge, which effects on the decrease of the heat source friction.

Figure 10a,b shows the relation between the average maximum temperature and feed rate during the applied cutting speed of *v_c_* = 60 m/min and *v_c_* = 100 m/min.

The analysis of Figure 10 shows that an applied lower cutting speed causes higher temperature values in the cutting zone. A similar correlation was observed for the components of the cutting forces.

## 6. Conclusions

The following can be concluded from the performed experimental research:Speed rate *f* has a significant effect on the values of the cutting forces. The speed rate increase causes the linear increase of all components of the cutting forces. The values of the cutting forces can be decreased by the increase of the cutting speed values. During the applied cutting speed of *v_c_* = 100 m/min, the total cutting force is about threefold lower than for the applied *v_c_* = 60 m/min.Surface roughness values (*Ra* and *Rz*) are connected to the feed rate *f* and cutting speed *v_c_*. For threefold increase of the speed rate *f*, values of surface roughness parameters *Ra* increase 2.5-fold, and values of surface roughness parameters *Rz* increase about 1.5-fold. In addition, higher values of surface roughness parameters (*Ra* and *Rz*) were obtained for *v_c_* = 100 m/min than *v_c_* = 60 m/min.Values of the average maximum temperature *T_max_* in the cutting zone decrease with the increase of the speed rate *f* and the cutting speed *v_c_*; the correlations are connected to chip thickness and contact time chip between the chip and the cutting edge, respectively.The applied cutting data have no effect on the chips form—all of them were unacceptable.

## Figures and Tables

**Figure 1 materials-12-00182-f001:**
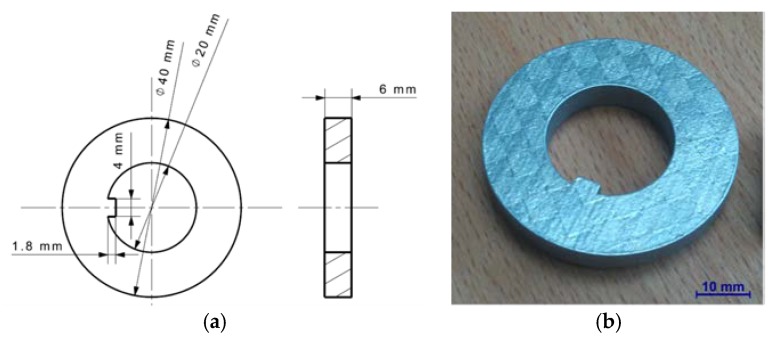
Test sample fabricated by using SLM: (**a**) dimensions of the sample; and (**b**) photograph of the sample.

**Figure 2 materials-12-00182-f002:**
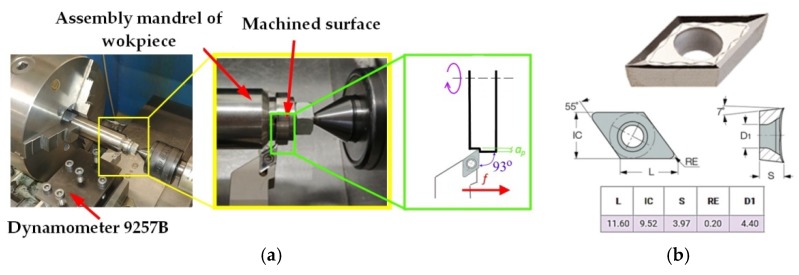
Test stand for experiments: (**a**) the attachment bolt of sample with dynamometer; and (**b**) dimensions of the DCGT 11T302-AS IC20 cutting insert.

**Figure 3 materials-12-00182-f003:**
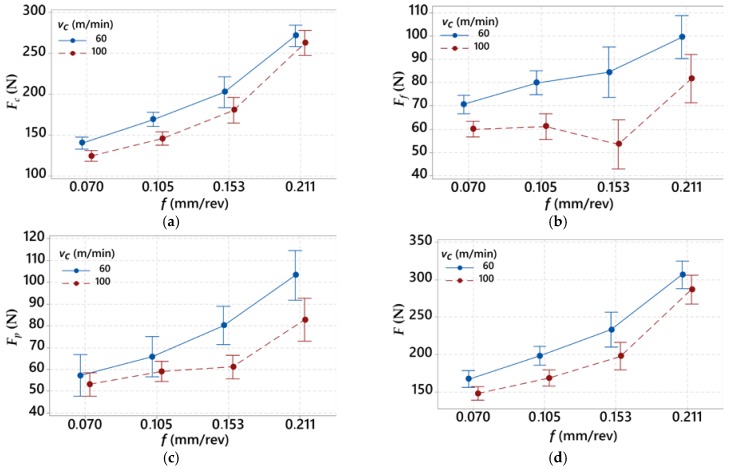
Diagram representing relation between the feed rate *f* and: (**a**) main force *F_c_*; (**b**) feed force *F_f_*; (**c**) thrust force *F_p_*; and (**d**) total cutting force *F*.

**Figure 4 materials-12-00182-f004:**
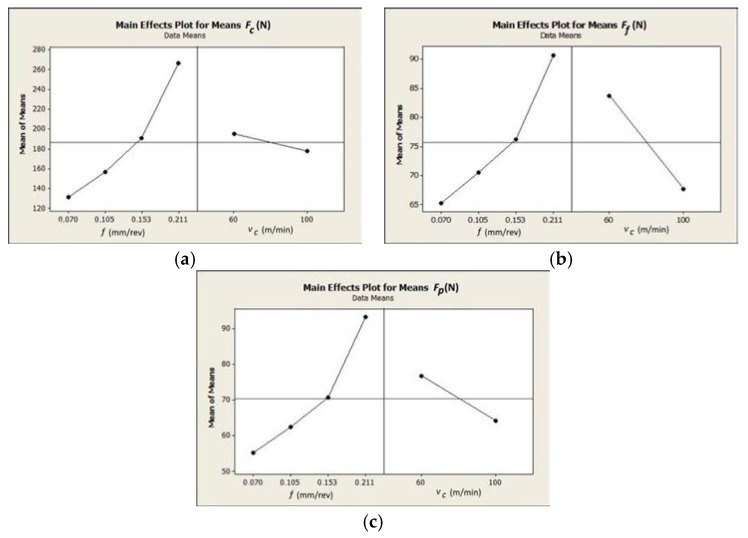
Diagram representing relation between the cutting data (*f*, *v_c_*) and: (**a**) main force *F_c_*; (**b**) feed force *F_f_*; and (**c**) thrust force *F_p_*.

**Figure 5 materials-12-00182-f005:**
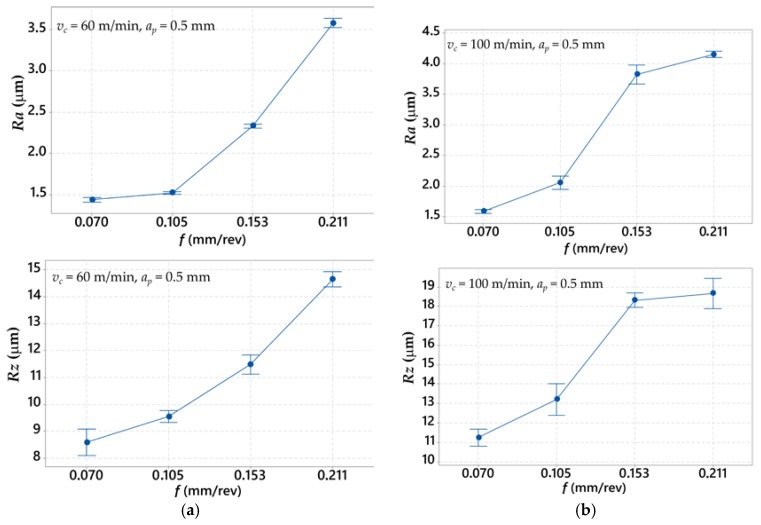
Graph representing relation between the average values of surface roughness parameters (*Ra* and *Rz*) and: (**a**) cutting speed *v_c_* = 60 m/min; and (**b**) cutting speed *v_c_* = 100 m/min.

**Figure 6 materials-12-00182-f006:**
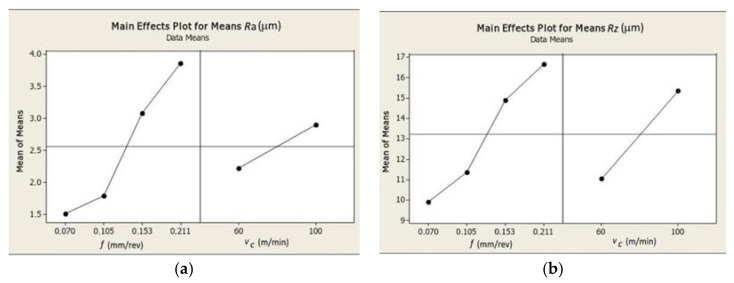
Graphs representing relation between the cutting data (*f*, *v_c_*) and: (**a**) surface roughness parameter *Ra*; and (**b**) surface roughness parameter *Rz*.

**Figure 7 materials-12-00182-f007:**
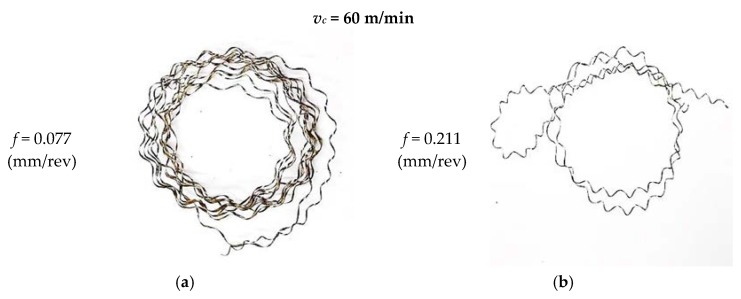
Photographs of example chips for *f_min_* = 0.07 mm/rev (**a**) and *f_max_* = 0.211 mm/rev (**b**).

**Figure 8 materials-12-00182-f008:**
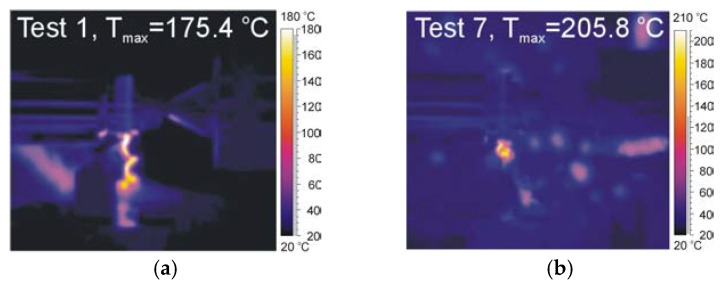
Thermograms recorded by ThermaCamResearcher software for: (**a**) Trial 1, *f_min_* = 0.07 mm/rev; and (**b**) Trial 7, *f_max_* = 0.211 mm/rev.

**Figure 9 materials-12-00182-f009:**
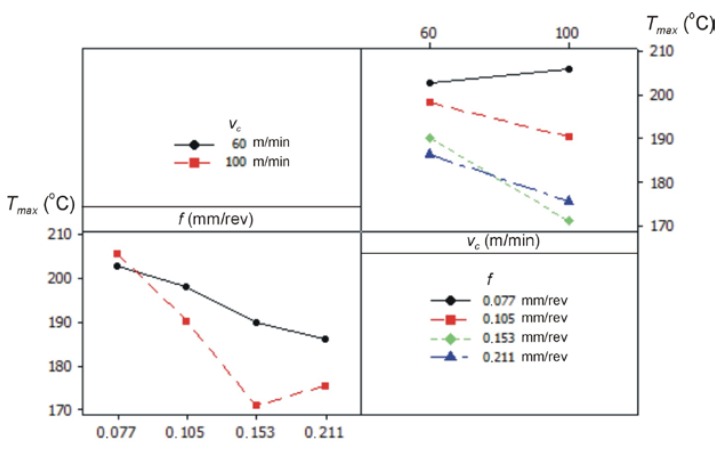
Graphs representing the relation between the average maximum temperature *T_max_* and the cutting data (*f*, *v_c_*).

**Figure 10 materials-12-00182-f010:**
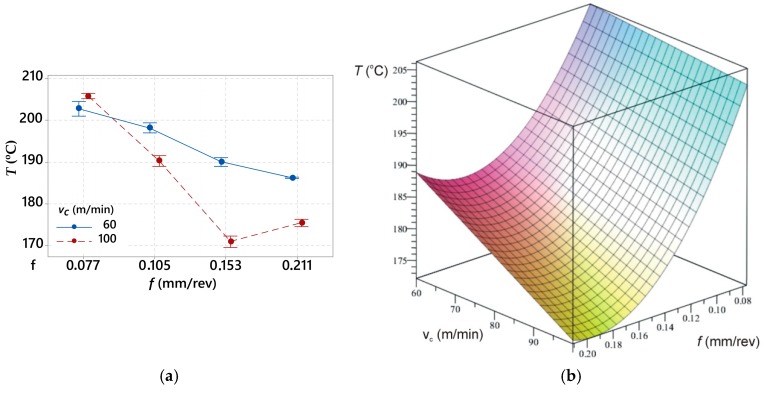
(**a**) Diagram representing relation between the average maximum temperature *T_max_* and the feed rate *f* with an applied cutting speed of *v_c_* = 60 m/min and *v_c_* = 100 m/min; and (**b**) 3D graph.

**Table 1 materials-12-00182-t001:** The major SLM process parameters used in printing 3D test samples.

Sintering Parameters	Value
layer thickness	20 µm
inert gas speed	2.5 m/s
laser power	90 W
laser speed	500 mm/s
coater work speed	80 mm/s
coater return speed	250 mm/s
oxygen level	0.3%

**Table 2 materials-12-00182-t002:** Chemical composition of 316L stainless steel (wt. %).

Symbol	C	Mn	Si	P	S	Cr	Mo	Ni	N
316L	0.03	2.0	0.75	0.045	0.03	17.0	2.7	13.0	0.1

**Table 3 materials-12-00182-t003:** Mechanical properties of 316L stainless steel.

Symbol	Tensile StrengthRm (MPa)	ElongationA5 (%)	Brinell Hardness HB	Thermal Conductivity @ 20 °C (W/(m × K))	Density (kg/dm^3^)	Yield Strength @ 20 °C Rp 0.2 (MPa)
316L	530–680	40	200	15	8.0	200

**Table 4 materials-12-00182-t004:** Experimental design by using the Taguchi method.

Trial	Feed Rate *f* (mm/rev)	Cutting Speed *v_c_* (m/min)	Depth of Cut *a_p_* (mm)
1	0.07	60	0.5
2	0.07	100	0.5
3	0.105	60	0.5
4	0.105	100	0.5
5	0.153	60	0.5
6	0.153	100	0.5
7	0.211	60	0.5
8	0.211	100	0.5

**Table 5 materials-12-00182-t005:** Experimental design and results of the components of the total cutting force.

Trial	*f* (mm/rev)	*v_c_* (m/min)	*F_c_* (N)	*Std. Dev. for F_c_*	*F_f_* (N)	*Std. Dev. for F_f_*	*F_p_* (N)	*Std. Dev. for F_p_*	*F* (N)	*Std. Dev. for F*
1	0.07	60	140	13	71	7	56	17	166	23
2	0.07	100	124	12	60	6	52	9	148	17
3	0.105	60	169	15	80	9	66	16	198	24
4	0.105	100	145	14	62	10	59	8	168	18
5	0.153	60	202	33	84	19	80	15	233	39
6	0.153	100	180	27	55	18	62	9	198	30
7	0.211	60	271	23	100	16	103	20	306	33
8	0.211	100	261	26	83	18	84	17	286	33

**Table 6 materials-12-00182-t006:** The results analysis according to the Taguchi method for the cutting forces *F_c_*, *F_f_*, and *F_p_*.

Trial	*f* (mm/rev)	*v_c_* (m/min)	*S/N__Fc_*	*F_c_mean_* (N)	*S/N__Ff_*	*F_f_mean_* (N)	*S/N__Fp_*	*F_p_mean_* (N)
1	0.07	60	−42.9	139.8	−37.0	70.6	−35.4	57.2
2	0.07	100	−41.9	123.9	−35.6	60.0	−34.6	53.2
3	0.105	60	−44.6	168.6	−38.1	80.0	−36.5	65.9
4	0.105	100	−43.3	145.3	−35.8	61.2	−35.5	59.1
5	0.153	60	−46.2	202.2	−38.7	84.5	−38.2	80.3
6	0.153	100	−45.2	180.1	−36.7	68.0	−35.8	61.2
7	0.211	60	−48.7	271.1	−40.0	99.6	−40.4	103.4
8	0.211	100	−48.4	262.3	−38.4	81.6	−38.5	82.9

**Table 7 materials-12-00182-t007:** Analysis of variance for mean values—main force *F_c_.*

Source	DF	Seq SS	Adj SS	Adj MS	*F*	*P*
*f*	3	20,630.4	20,630.4	6876.81	315.87	0.000
*v_c_*	1	612.5	612.5	612.50	28.13	0.013
Residual Error	3	65.3	65.3	21.77		
Total	7	21,308.2				

**Table 8 materials-12-00182-t008:** Analysis of variance for mean values—feed force *F_f_*.

Source	DF	Seq SS	Adj SS	Adj MS	*F*	*P*
*f*	3	712.91	712.91	237.638	35.13	0.008
*v_c_*	1	509.34	509.34	509.337	75.29	0.003
Residual Error	3	20.29	20.29	6.765	-	-
Total	7	1242.54	-	-	-	-

**Table 9 materials-12-00182-t009:** Analysis of variance for mean values—thrust force *F_p_*.

Source	DF	Seq SS	Adj SS	Adj MS	*F*	*P*
*f*	3	1620.2	1620.2	540.06	15.34	0.025
*v_c_*	1	317.5	317.5	317.52	9.02	0.058
Residual Error	3	105.6	105.6	35.21	-	-
Total	7	2043.3	-	-	-	-

**Table 10 materials-12-00182-t010:** The results of surface roughness *Ra* and *Rz*.

Trial	*f* (mm/rev)	*v_c_* (mm/min)	*Ra* (µm)	*Std. Dev. for Ra*	*Rz* (µm)	*Std. Dev. for Rz*
1	0.07	60	1.44	0.05	8.58	0.84
2	0.07	100	1.58	0.06	11.24	0.76
3	0.105	60	1.52	0.03	9.54	0.36
4	0.105	100	2.05	0.19	13.20	1.43
5	0.153	60	2.33	0.04	11.48	0.62
6	0.153	100	3.83	0.26	18.33	0.62
7	0.211	60	3.58	0.10	14.65	0.50
8	0.211	100	4.15	0.09	18.67	1.33

**Table 11 materials-12-00182-t011:** The examples of topography and profiles of the parts surface for *f_min_* = 0.07 mm/rev and *f_max_* = 0.211 mm/rev, *v_c_* = 60 m/min.

Trial	*f*	*v_c_*	Surface Topography	Surface Profile
	(mm/rev)	(m/min)		
1	0.07	60	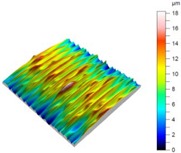	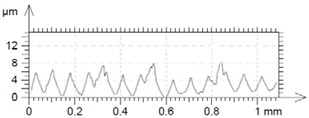
7	0.211	60	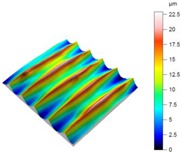	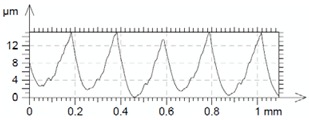

**Table 12 materials-12-00182-t012:** The results analysis of S/N factor and surface roughness parameters (*Ra*, *Rz*) according to Taguchi method.

Trial	*f* (mm/rev)	*v_c_* (m/min)	*S/N__Ra_*	*Ra* (µm)	*S/N__Rz_*	*Rz* (µm)
1	0.07	60	−3.15	1.44	−18.69	8.58
2	0.07	100	−4.00	1.58	−21.03	11.24
3	0.105	60	−3.64	1.52	−19.60	9.54
4	0.105	100	−6.27	2.05	−22.45	13.20
5	0.153	60	−7.35	2.33	−21.21	11.48
6	0.153	100	−11.67	3.83	−25.27	18.33
7	0.211	60	−11.08	3.58	−23.32	14.65
8	0.211	100	−12.37	4.15	−25.44	18.67

**Table 13 materials-12-00182-t013:** Analysis of variance for average values of the surface roughness parameter *Ra*.

Source	DF	Seq SS	Adj SS	Adj MS	*F*	*P*
*f*	3	7.353	7.353	2.4511	14.94	0.026
*v_c_*	1	0.945	0.945	0.9453	5.76	0.096
Residual Error	3	0.492	0.492	0.164		
Total	7	8.7905				

**Table 14 materials-12-00182-t014:** Analysis of variance for average values of surface roughness parameter *Rz*.

Source	DF	Seq SS	Adj SS	Adj MS	*F*	*P*
*f*	3	58.10	58.101	19.367	12.06	0.035
*v_c -_*	1	36.937	36.937	36.937	23.01	0.017
Residual Error	3	4.817	4.817	1.606		
Total	7	99.855				

**Table 15 materials-12-00182-t015:** The configuration parameters of the thermal camera.

Flir SC 620	Parameter	Value
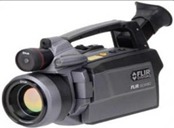	ambient temperature	22 °C
	emissivity	0.98
	relative humidity	37%
	distance measurement	1 m
	image frequency	30 frame/s

**Table 16 materials-12-00182-t016:** The statistical analysis results of the maximum temperature *T_max_* in the cutting zone.

Trial	*f* (mm/rev)	*v_c_* (m/min)	*S/N* (dB)	*T_max_* (°C)	*Std. Dev.*
1	0.211	100	−44.9	175.4	1.1
2	0.211	60	−45.4	186.2	3.1
3	0.153	100	−44.6	170.9	2.2
4	0.153	60	−45.6	190.0	2.1
5	0.105	100	−45.6	190.3	2.3
6	0.105	60	−45.9	198.2	1.9
7	0.077	100	−46.3	205.8	1.5
8	0.077	60	−46.1	202.8	0.4

**Table 17 materials-12-00182-t017:** Analysis of variance for average values of the maximum temperature—*T_max_*.

Source	DF	SeqSS	Adj SS	Adj MS	*F*	*P*
*f*	3	796.8	796.8	265.59	6.37	0.081
*v_c_*	1	151.4	151.4	151.38	3.63	0.153
Residual Error	3	125.0	125.0	41.68	-	-
Total	7	1073.2	-	-	-	-

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
