# Peer review of "Turning Research of Additive Laser Molten Stainless Steel 316L Obtained by 3D Printing"

_materials, 2019, doi:10.3390/ma12010182_

Round 1
Reviewer 1 Report
The work presented in this manuscript is interesting, well written and explained. Nonetheless some changes, correction and clarifications must be done to be, in my opinion, publishable.
Second page, 2.1 section. It is not clear hoy you are machining your part. I guess that you are reducing the original sample thickness from 6 mm to something smaller. You are facing the part? If this is the case make it clear in the text. Figure 2(a) show a turning and this is not the operation made.
Figure 2. Shows parameters related with the cutting insert but the cutting angle “Tao” is not shown in the table.
Table 1 must be placed, if possible after figure 1.
Line 92. “Formed chips are subject to estimate”. What do you mean with estimate… complete the sentence or change it for another one explain what you are going to study about chip geometry.
Line 97. What acquisition hardware you use for digitalizing the electrical signal obtained from the amplifier? What acquisition conditions (Sampling frequency and bit depth)?
Line 100… the “hardness average value”… means “average hardness value”?
Line 102. HRC_mean, write the mean as subscript.
Table 3… “Thermal conductivity in temperature 20 (ºC)”… write it as “Thermal conductivity @20ºC”.
Table 2 and 3. In the test you named the material used as 316L (line 105) in these tables you used a 316l replace the “l” with “L”.
Line 116. Give any reference to the Taguchi method. You use it through all the manuscript and do not give any reference about it.
Line 119. Explain in the text what each force means. Perhaps a new figure/scheme with the tool, the part and axis will help a lot.
Line 124 you talk about a linear polynomial…. It it is linear it is linear if it’s a polynomial is not linear… keep only the polynomial term.
Table 4 and others. You use nomenclature or parameters A and B. But you do not explain them in the text. It is clear that if a way of controlling/handling with your tests, but you do not mention on the text and in some table you use them an in others no. I think the best is to remove all of them.
Line 126. Explain more what the s/N parameter means and how you calculate it, what y_i and n means… You just show eq (1) and do not say anything about its meaning or about what variables you use in that expression.
Table 5. Include the total force. I know that you can calculate if with the Fc, Ff and Fp but it is a value that you plot later in figure 3.
Table 5. Can you give any estimation about the error that you have measuring the forces?
Figure 3. The way of showing units is no correct. You use, for example, “Fc N”. Use “Fc [N]”, “Fc/N” of even “Fc (N)” but not “Fc N”. The same with all axis in all figures
Figure 3. If you can give estimate in table 5 plot errorbars.
Line 146. The end of the sentence is no clear. I suggest changing the end of the sentence with “the use of higher cutting speeds (100 m/min vs 60 m/min).”
Table 6. Idem as table 5 (errors)
Figure 4. Put on f anc vc on plots the units.
Table 11. In right-side plots I suggest using the same vertical scale, i.e. from 0 to 16 to make comparison between both condition easier.
Line 186. What you mean with proportionate? Proportional?
Page 10 section 5. You do not give information about how you measure the temperature. I mean you recorded the complete machining process and gets the maximum temperature of the complete sequence? Or only a pixel of the image. You get info only from a single frame… the average of maximum along time?? Give more information about it, please.
Table 16. If you average temperature along time… can you estimate any error in temp. measurement?
Line 231. Lineal polynomial. You have f^2 so it is a polynomial fitting.
Line 261, 262. Use 1.5 and 2.5 instead of one-and-a-half.
Author Response
The work presented in this manuscript is interesting, well written and explained. Nonetheless some changes, correction and clarifications must be done to be, in my opinion, publishable.
Thank you!
Second page, 2.1 section. It is not clear how you are machining your part. I guess that you are reducing the original sample thickness from 6 mm to something smaller. You are facing the part? If this is the case make it clear in the text. Figure 2(a) show a turning and this is not the operation made.
No, it is longitudinal turning on a distance of 6 mm. Figure 2 was changed to show the turning process more clear.
Figure 2. Shows parameters related with the cutting insert but the cutting angle “Tao” is not shown in the table.
The angle between the cutting edge and feed direction is 93o.
Table 1 must be placed, if possible after figure 1.
Table 1 was placed after figure 1.
Line 92. “Formed chips are subject to estimate”. What do you mean with estimate… complete the sentence or change it for another one explain what you are going to study about chip geometry.
Explanation of the classification of the chip formation (advantages, disadvantages, acceptable or unacceptable) was added.
Line 97. What acquisition hardware you use for digitalizing the electrical signal obtained from the amplifier? What acquisition conditions (Sampling frequency and bit depth)?
The cutting forces waveforms were convert by analog to digital converter and analyzed by using DynoWare software. Sampling frequency was 1 kHz and measuring time 10 s.
Line 100… the “hardness average value”… means “average hardness value”?
It was changed to average hardness value.
Line 102. HRC_mean, write the mean as subscript.
It was write as subscript.
Table 3… “Thermal conductivity in temperature 20 (ºC)”… write it as “Thermal conductivity @20ºC”.
It was changed.
Table 2 and 3. In the test you named the material used as 316L (line 105) in these tables you used a 316l replace the “l” with “L”.
It was replaced.
Line 116. Give any reference to the Taguchi method. You use it through all the manuscript and do not give any reference about it.
The following reference was added: Nalbant, T.M.; Gökkaya, H.; Sur, G. Application of Taguchi method in the optimization of cutting parameters for surface roughness in turning. Materials and Design 2007, 28, 1379–1385.
Line 119. Explain in the text what each force means. Perhaps a new figure/scheme with the tool, the part and axis will help a lot.
The explanation what the force components mean was included to the paper text. There are main force Fc, feed force Ff and thrust force Fp.
Line 124 you talk about a linear polynomial…. It it is linear it is linear if it’s a polynomial is not linear… keep only the polynomial term.
It was changed, so only polynomial term is used in the text of the paper.
Table 4 and others. You use nomenclature or parameters A and B. But you do not explain them in the text. It is clear that if a way of controlling/handling with your tests, but you do not mention on the text and in some table you use them an in others no. I think the best is to remove all of them.
Parameters A and B were removed from all tables.
Line 126. Explain more what the s/N parameter means and how you calculate it, what y_i and n means… You just show eq (1) and do not say anything about its meaning or about what variables you use in that expression.
The explanation of the S/N, yi and n parameters were included to the paper text.
Table 5. Include the total force. I know that you can calculate if with the Fc, Ff and Fp but it is a value that you plot later in figure 3.
The total force F was included.
Table 5. Can you give any estimation about the error that you have measuring the forces?
The estimated errors were added to the table.
Figure 3. The way of showing units is no correct. You use, for example, “Fc N”. Use “Fc [N]”, “Fc/N” of even “Fc (N)” but not “Fc N”. The same with all axis in all figures
It was changed to “( )” in all figures.
Figure 3. If you can give estimate in table 5 plot error bars.
The error bars were added.
Line 146. The end of the sentence is no clear. I suggest changing the end of the sentence with “the use of higher cutting speeds (100 m/min vs 60 m/min).”
It was changed.
Table 6. Idem as table 5 (errors)
The estimated errors were added to other tables.
Figure 4. Put on f anc vc on plots the units.
The units were put on plots.
Table 11. In right-side plots I suggest using the same vertical scale, i.e. from 0 to 16 to make comparison between both condition easier.
The same vertical scale was used for both plots.
Line 186. What you mean with proportionate? Proportional?
It was changed to proportional.
Page 10 section 5. You do not give information about how you measure the temperature. I mean you recorded the complete machining process and gets the maximum temperature of the complete sequence? Or only a pixel of the image. You get info only from a single frame… the average of maximum along time?? Give more information about it, please.
Yes, we recorded the 2 second sequences of a stable phase of machining process (30 frames per second) and got the maximum temperature which existed in the cutting area.
Table 16. If you average temperature along time… can you estimate any error in temp. measurement?
The main errors during temperature measurements are the emissivity factor and reflections originating from the lighting of the research stand. In our case the emissivity factor was 0.98.
Line 231. Lineal polynomial. You have f^2 so it is a polynomial fitting.
It was changed to polynomial.
Line 261, 262. Use 1.5 and 2.5 instead of one-and-a-half.
It was changed to 1.5 fold and 2.5 fold.
Reviewer 2 Report
In this manuscript, the authors studied turning research of additive laser molten stainless steel 316L obtained by 3D printing. I have the following comments:
1- There is no point in including all the 3D printing/ additive manufacturing processes in the introduction section.
2- Can the authors comments about the differences between the machining behavior/ properties of 3D printing parts vs. machining parts.
3- There are a great deal of recently published literature concerning 3D printing of 316L. The authors need to go through the following example of recently published work and cite it in order to relate their findings with previously published work:
- Investigation into spatter behavior during selective laser melting of AISI 316L stainless steel powder
- Thermal behavior of the molten pool, microstructural evolution, and tribological performance during selective laser melting of TiC/316L stainless steel nanocomposites: Experimental and simulation methods
- Parametric analysis of thermal behavior during selective laser melting additive manufacturing of aluminum alloy powder
- Selective laser melting of TiC reinforced 316L stainless steel matrix nanocomposites: Influence of starting TiC particle size and volume content
Author Response
In this manuscript, the authors studied turning research of additive laser molten stainless steel 316L obtained by 3D printing. I have the following comments:
1- There is no point in including all the 3D printing/ additive manufacturing processes in the introduction section.
The part of text including various 3D printing processes was removed (line 37-44).
2- Can the authors comments about the differences between the machining behavior/ properties of 3D printing parts vs. machining parts.
The following comments were included to the Introduction section:
Both subtractive and additive manufacturing have a number of advantages and disadvantages. In practice, parts made in additive technology are a replacement for casting technology. The big advantages of the additive manufacturing (AM) rely basically on no restrictions to building complex shapes and ability to produce several units at the same time. On the other hand, the disadvantage is undoubtedly the lower surface quality and the dimensional - shape accuracy compared to the parts made with the use of machining. The material obtained with 3D printing technology shows porosity and areas with different consistency of material. The characteristics of the parts obtained in the AM technology is the layered structure of the material and necessity to decrease the size of stresses within the material caused by a poor heat distribution [Rev1,Rev2].
Geometrically simple objects are easier to machine by SM. A more complex tool path is generated if the geometry becomes more complex and this can be difficult to produce with a milling machine even with five or more axes.
An interesting idea to improve the quality of the manufactured parts is the use of hybrid processing, previously analyzing whether it is better to use more thin layers during AM production or less (then worse quality of the item) but with the need of additional mechanical machining AM + SM [Rev3].
Rev1. Kruth JP, Levy G, Klocke F, Childs THC. Consolidation phenomena in laser and powder-bed based layered manufacturing. CIRP Ann Manuf Technol 2007;56:730–59.
Rev2. Al Jabbari YS, Koutsoukis T, Barmpagadaki X, Zinelis S. Metallurgical and interfacial characterization of PFM Co–Cr dental alloys fabricated via casting, milling or selective laser melting. Dent Mater 2014;30:e79.
Rev3. Fernández M, Delgado L, Molmeneu M, García D, Rodríguez D. Analysis of the misfit of dental implant-supported prostheses made with three manufacturing processes. JProsthet Dent 2014;111:116–23.
3- There are a great deal of recently published literature concerning 3D printing of 316L. The authors need to go through the following example of recently published work and cite it in order to relate their findings with previously published work:
- Investigation into spatter behavior during selective laser melting of AISI 316L stainless steel powder
- Thermal behavior of the molten pool, microstructural evolution, and tribological performance during selective laser melting of TiC/316L stainless steel nanocomposites: Experimental and simulation methods
- Parametric analysis of thermal behavior during selective laser melting additive manufacturing of aluminum alloy powder
- Selective laser melting of TiC reinforced 316L stainless steel matrix nanocomposites: Influence of starting TiC particle size and volume content
The following text was included to the Introduction section:
The effect of LSM parameters like laser power, scanning velocity and scanning direction are correlated with the mechanical properties of the fabricated parts [Rev6,8,9]. Applied lower laser power (100 W) causes that fabricated sample characterizes more pores and lower hardness [Rev9]. However, the lower laser power (200 W) and the high scan speed (400 mm/s) cause yielded a low temperature during process SLM what gives poor wettability and appearing micro-pores in structural [Rev6]. Additively, the laser power increase and or scanning velocity decrease contribute to the increase of energy density which impact on: lower porosity, bigger melt pool and stronger bonding between pools [Rev8]. The quality of fabricated part depends on these properties which can effect on the machining post–process.
Additively, the parameter of SLM such as energy input effects spatter behavior which is generated during the process. The formed spatter influence on the part structure what can be significant during added machining post–process [Rev4]. The distribution state and particle size of nanoscale TiC reinforcements in the structure of molten part, effect on the micro-hardness and wear rates [Rev5]. The volume content of TiC increase causes hardness increase and coefficient of friction and wear rate [Rev7].
Rev4. Liu, Y.; Yang Y.; Mai, S.; Wang, D.; Song, C. Investigation into spatter behavior during selective laser melting of AISI 316L stainless steel powder. Materials & Design 2015, 87, 797-806.
Rev5. AlMangour, B.; Grzesiak, D.; Cheng, J.; Ertas, Y. Thermal behavior of the molten pool, microstructural evolution, and tribological performance during selective laser melting of TiC/316L stainless steel nanocomposites: Experimental and simulation methods. Journal of Materials Processing Technology 2018, 257, 288-301.
Rev6. Li, Y.; Gu, D. Parametric analysis of thermal behavior during selective laser melting additive manufacturing of aluminum alloy powder. Materials & Design 2014, 63, 856-867.
Rev7. AlMangour, B.; Grzesiak, D.; MingYang, J. Selective laser melting of TiC reinforced 316L stainless steel matrix nanocomposites: Influence of starting TiC particle size and volume content. Materials & Design 2016, 104, 141-151..
Rev8. Ahmadi, A.; Mirzaeifar, R.; Moghaddam, N. S.; Turabi, A. S.; Karaca, H. E.; Elahinia, M. Effect of manufacturing parameters on mechanical properties of 316L stainless steel parts fabricated by selective laser melting: A computational framework. Materials and Design 2016, 112, 328–338.
Rev9. Li, H.; Ramezani, M.; Li M.; Ma, Ch.; Wang, J. Effect of process parameters on tribological performance of 316L stainless steel parts fabricated by selective laser melting. Manufacturing Letters 2018, 16, 36–39.